# Epigenetic Landscapes of Aging in Breast Cancer Survivors: Unraveling the Impact of Therapeutic Interventions—A Scoping Review

**DOI:** 10.3390/cancers17050866

**Published:** 2025-03-03

**Authors:** Nikita Nikita, Zhengyang Sun, Swapnil Sharma, Amy Shaver, Victoria Seewaldt, Grace Lu-Yao

**Affiliations:** 1Department of Medical Oncology, Sidney Kimmel Medical College, Thomas Jefferson University, Philadelphia, PA 19107, USA; fnu.nikita@jefferson.edu (N.N.); zhengyang.sun@students.jefferson.edu (Z.S.); swapnil.sharma@jefferson.edu (S.S.);; 2Sidney Kimmel Comprehensive Cancer Center, Thomas Jefferson University, Philadelphia, PA 19107, USA; 3Department of Pharmacology, Physiology and Cancer Biology, Sidney Kimmel Medical College, Thomas Jefferson University, Philadelphia, PA 19107, USA; 4Department of Population Sciences, City of Hope Comprehensive Cancer Center, Duarte, CA 91010, USA; vseewaldt@coh.org

**Keywords:** breast cancer, cancer therapy, cancer survivorship, aging, epigenetics, biological aging, DNA methylation, histone modifications, non-coding RNA, microRNA, therapy-induced aging

## Abstract

While breast cancer treatments have significantly improved survival rates, they also bring long-term health considerations, especially for aging survivors. This scoping review investigates how breast cancer therapies intersect with aging, emphasizing the epigenetic changes induced by chemotherapy and targeted therapies. These treatments often speed up biological aging through DNA methylation, histone modifications, and chromatin remodeling. This review discusses the dual role of epigenetic modifications in breast cancer therapy, considering both the treatment benefits and their potential contribution to biological aging. It examines epigenetic landscapes first from a microscopic perspective, including DNA and histone methylations, chronic inflammation, telomere shortening, and non-coding RNA modifications. It also discusses the macroscopic epigenetic-related clinical manifestations of aging-related adverse effects, such as cardiovascular, respiratory, and endocrine dysfunctions, highlighting the need for tailored long-term care and multidisciplinary approaches. Additionally, emerging therapeutic strategies aimed at these epigenetic modifications are considered for their potential to mitigate therapy-induced aging. Understanding these epigenetic landscapes is crucial for enhancing breast cancer survivors’ health span and quality of life. By bridging molecular insights with clinical implications, this review underscores the urgent need for integrative approaches and research that address therapy-induced aging in breast cancer survivors.

## 1. Introduction

In 2024, approximately 313,510 patients will develop breast cancer in the United States, with an estimated annual death of 42,250 [1]. With recent innovations in breast cancer therapies, including chemotherapy, radiation, hormonal treatments, and targeted therapies, the 5-year relative survival rates have been constantly improving, from 75% in 1975–1977 to 91% in 2013–2019 [1]. This success in improving treatment regimens not only provided life and hope for breast cancer patients, but also created an urgent need to provide appropriate long-term care for the aging population of breast cancer survivors. There are currently about 2.5 million breast cancer survivors in the U.S., and many of them are over the age of 65 [2,3]. Simultaneously, the occurrence of breast cancer is intimately associated with age, with an increasing incident rate from 1 in 48 in females aged 49 or younger to 1 in 14 in females aged 69 to 84 years old, which may lead to an increasing public health and societal burden due to the aging of the general population [1]. Therefore, the complexity between aging and breast cancer survivorship requires adequate health outcome research to provide tailored care and improve patients’ quality of life (QoL).

Biological aging is generally defined as accumulated damage to biological systems over the life course, causing morbidity and disability [4]. Importantly, biological age is not always immediately apparent when the patient presents for oncology care but may be indirectly reflected in comorbidities, polypharmacy, and functional limitations [5]. Biological age needs to be considered, particularly since many of our standard-of-care and next-generation therapeutics may inadvertently expedite biological aging and result in increased senescence, age-related changes, and a higher incidence of age-associated diseases [6,7].

Biological aging is influenced by epigenetic modifications, encompassing DNA methylation, histone modifications, and chromatin remodeling. These epigenetic modifications affect gene expression without altering the underlying DNA sequence, contributing to the aging phenotype [8,9]. Moreover, exploring epigenetic biomarkers of aging, such as the epigenetic clock, provides a compelling avenue for assessing the biological age of cells and tissues. This approach offers valuable insights into the cumulative impact of cancer therapy on the aging process [10]. It is worth noting that novel therapeutic interventions targeting epigenetic modifications are emerging as potential strategies to mitigate therapy-induced aging. The goal of these interventions is to enhance both the lifespan and health span of breast cancer survivors [11].

In breast cancer survivors, the interface between cancer treatment and aging has gained significant attention in the scientific community. The study of epigenetic modification has become a crucial area of investigation due to its profound impact on long-term patient outcomes and overall quality of life (QoL). Recent research findings have been synthesized through a scoping review to comprehensively understand the effects of various cancer therapies on aging. This review aims to shed light on the intricate relationship between breast cancer treatments and the aging process, with a particular focus on the epigenetic alterations that contribute to the aging phenomena found among breast cancer survivors.

An emerging concern is how these epigenetic alterations may converge with lifestyle factors—such as diet, physical activity, and stress—to further exacerbate or mitigate accelerated aging. Breast cancer survivors often undergo extensive therapeutic regimens that can lead to prolonged periods of elevated systemic inflammation, oxidative stress, and hormonal dysregulation, each of which is itself associated with changes in the epigenetic landscape. By incorporating geriatric assessments, clinicians can better measure vulnerability to therapy-related toxicities and age-related morbidities, guiding a more personalized treatment plan. This integrative approach can improve not only survival but also the functional status and quality of life of older breast cancer survivors, acknowledging the cumulative burden of cancer therapy and aging [12,13].

## 2. Cancer Therapy’s Impact on Aging-Related Epigenetic Pathways

The impact on aging-related pathways has been extensively studied in cancer research. Researchers have found that various mechanisms, such as DNA methylation, histone modifications, and epigenetically active drugs, can play a crucial role in differentiating cancer cells, assessing risk, and improving treatment outcomes [12]. These epigenetic factors, along with changes in the chromatin landscape, have been implicated in therapeutic resistance in breast cancer [13].

Specifically, the administration of chemotherapy has been shown to induce a state of cellular senescence, characterized by the upregulation of senescence-associated markers and epigenetic modifications. Chemotherapy may target cancer cells and induce systemic aging effects throughout the body. The consequences of chemotherapy on aging-related pathways extend beyond the tumor microenvironment and have broader implications for the overall health and well-being of cancer patients [14]. Similarly, while focusing on specific cellular pathways as the main indication, targeted therapy for breast cancer can also influence epigenetic regulation. In recent years, epigenetic-based therapies have gained attention as potential strategies to counteract the adverse epigenetic impacts of conventional treatments. These innovative therapies aim to reverse DNA methylation and histone modifications, which are key players in epigenetic regulation [15].

Moreover, there is growing evidence that cancer-associated fibroblasts (CAFs) within the breast tumor microenvironment can acquire senescence phenotypes, further driving tumor progression and systemic aging. CAFs undergo significant epigenetic changes during therapy, producing senescence-associated secretory phenotype (SASP) factors that amplify inflammation and tissue remodeling. These alterations underscore how not only tumor cells but also the surrounding stromal cells may contribute to therapy-induced aging, thereby highlighting the importance of a comprehensive, multi-cellular perspective on breast cancer treatment [14,15].

### 2.1. DNA Methylation in Cancer Therapy

DNA methylation, a critical epigenetic mechanism, assumes a pivotal role in suppressing tumor suppressor genes and activating oncogenes, profoundly impacting cancer development and progression. For cancer therapy, targeted interventions aimed at modulating DNA methylation have exhibited immense potential in reactivating genes that had been silenced and sensitizing cancer cells to various treatment modalities, including breast cancer treatment. 

In breast cancer, aberrant patterns of DNA methylation are known to culminate in the silencing of tumor suppressor genes, significantly contributing to the progression of this malignant disease. Consequently, the advent of epigenetic therapies that are specifically designed to target these modifications, such as DNA methyltransferase inhibitors (DNMTis), represents a highly promising approach to counteract gene silencing. Notably, numerous studies have diligently explored the therapeutic potential of DNMTis in conjunction with histone deacetylase inhibitors (HDACis) in reactivating silenced genes, thereby offering a highly encouraging and innovative strategy to enhance the sensitivity of cancer cells to chemotherapy and other treatment modalities [15]. Importantly, DNMTi-based therapies have also been linked with a reduced expression of senescence markers in pre-clinical models, hinting at a possible role in ameliorating therapy-induced aging [16].

Studies of patients undergoing breast cancer therapy have observed significant alterations in DNA methylation patterns, especially those associated with the acceleration of the epigenetic clock [16]. This implies that current breast cancer treatments may elicit cellular changes mimicking the aging process, potentially contributing to the early onset of age-related diseases in cancer survivors. This finding highlights the potential of DNA methylation profiles as biomarkers for monitoring biological aging due to cancer therapy.

Epigenetic drift refers to the stochastic changes in DNA methylation patterns observed with aging, which can influence cancer development. This drift contributes to epigenetic mosaicism in aging cells, potentially affecting cellular plasticity and leading to various age-related phenotypes and diseases. Epigenetic drift, characterized by stochastic changes in DNA methylation over time, contributes to cellular aging and can influence the development of cancer. Understanding the relationship between epigenetic drift and aging could provide insights into cancer prevention and the development of age-related diseases [17]. In addition to these findings, a systematic review found altered expression of TET (ten–eleven translocation) enzymes and levels of the 5-Hydroxymethylcytosine (5hmC) epigenetic mark in breast cancer, emphasizing potential therapeutic targets. The relationship between cancer therapy and biological aging indicators (DNA damage, reduced telomerase activity, and epigenetic aging) underscores the complexity of therapy-induced aging in breast cancer survivors.

Recent epigenome-wide association studies (EWASs) in breast cancer survivors further highlight distinct methylation signatures correlated with chemotherapeutic regimen exposure and radiotherapeutic dose, underscoring the specificity of epigenetic alterations to different treatment modalities [17,18].

### 2.2. Histone Modifications

Histone modifications, such as changes in macroH2A1, which serves as a marker of both senescence and cancer, have been intricately associated with chemotherapy-induced senescence, thus indicating a highly complex and intertwined relationship between epigenetic markers and response to therapy. It has been observed that breast cancer treatments can potentially induce modifications in histone acetylation and methylation, consequently impacting the expression patterns of critical genes involved in crucial cellular processes, including DNA repair, regulation of the cell cycle, and apoptosis. These alterations in the epigenetic landscape have the potential to promote cellular senescence, a state characterized by a permanent halt in the cell cycle, thereby playing a significant role in the aging process as well as age-related diseases. Remarkably, it has been established that chemotherapy can instigate the induction of the senescence-associated secretory phenotype (SASP), which manifests as the secretion of a wide array of pro-inflammatory cytokines, chemokines, and growth factors, ultimately fostering a pro-aging milieu within the cellular microenvironment [18].

**Role of Histone Modifications**: Histone modifications, in conjunction with DNA methylation, are crucial when it comes to the regulation of gene expression. Epigenetic alterations in histone acetylation and methylation patterns have the potential to disrupt critical pathways in breast cancer. Thus, comprehending these changes not only provides valuable insights into the disease but also presents opportunities for the identification of therapeutic targets and biomarkers that can aid in cancer progression and treatment responses. In the context of breast cancer, modifications in histone patterns can lead to the dysregulation of genes that are pivotal in controlling the cell cycle, apoptosis, and metastasis. As a result of studying these modifications, researchers have developed histone deacetylase inhibitors (HDACis), which have shown the ability to induce cancer cell death, reduce proliferation rates, and inhibit tumor growth. These findings underscore the significance of investigating the dynamics of histone modifications and highlight the potential therapeutic benefits of targeting epigenetic regulators in the treatment of breast cancer [9]. Furthermore, targeting histone reader proteins (e.g., bromodomain and extra-terminal domain (BET) family members) has emerged as a novel strategy to modulate aberrant transcriptional programs in breast cancer while potentially mitigating age-related inflammatory responses [19].

Beyond HDACs, emerging attention is directed towards histone methyltransferases (HMTs) and demethylases (KDMs) involved in the tri-methylation of histone H3 lysine residues (e.g., H3K27me3, H3K4me3). Dysregulation of these marks has been noted in multiple breast cancer subtypes and linked to heightened senescence pathways, reinforcing the connection between histone modifications and cellular aging [20].

**Telomere Length:** Alhareeri et al. reported an overall decrease in telomere length in breast cancer patients receiving chemotherapy. Telomere length was associated with self-perceived pain levels, elucidating potential mechanisms of epigenetic modification leading to psychomotor symptoms [19]. Interestingly, telomere length has also been found to be associated with general breast cancer prognosis. Specifically, shorter telomeres have been associated with older age, higher local recurrence rates, higher tumor burden, and lower physical activity and appear to be independent of hormonal receptor status [20]. Further investigations using long-term cohorts of breast cancer survivors have begun to integrate telomere length with epigenetic aging markers (such as DNA methylation clocks), offering a more comprehensive “multi-omic” approach to understanding how therapy accelerates biological aging [21].

**Chronic Inflammation:** The sequelae of cancer treatment may increase systemic inflammation and create a phenotype at increased risk of functional decline and comorbidities, leading to premature mortality. Alfano et al. reported that, over time, breast cancer survivors had significantly higher tumor necrosis factor-α and IL-6 compared to the control group, despite having no differences at the baseline level [21]. Survivors treated with surgery, radiation, and chemotherapy accumulated a significantly greater burden of comorbid conditions and suffered greater pain associated with inflammation over time after cancer treatment than the control group [22]. Recent studies further indicate that the inflammatory environment can mediate epigenetic changes, including altered histone acetylation and DNA methylation patterns, thereby creating a feedback loop which perpetuates both cellular senescence and chronic disease risk in survivors [14]. Immunosenescence, characterized by a decline in adaptive immune function and an increase in inflammatory phenotypes, has also been reported in aging breast cancer survivors. Altered epigenetic regulation in T-lymphocytes, such as changes in the methylation of key immune-related genes, has been associated with compromised immune surveillance and higher susceptibility to infections and secondary malignancies [22].

**Non-coding RNAs and Aging Pathways:** Non-coding RNAs, including microRNAs (miRNAs) and long non-coding RNAs (lncRNAs), are key regulators of gene expression that can be affected by breast cancer treatments. These molecules are involved in numerous cellular processes, including those related to aging, such as cellular senescence, DNA damage response, and mitochondrial function. Epigenetic therapies targeting miRNAs and lncRNAs offer a promising approach to modulating aging pathways altered by breast cancer treatments, potentially mitigating therapy-induced aging effects.

**MicroRNA and Epigenetics:** MicroRNAs (miRNAs) are small non-coding RNA molecules that play a crucial role in regulating gene expression post-transcriptionally. Their involvement in cancer and aging is significant, as the dysregulation of miRNAs can lead to alterations in the epigenetic landscape of breast cancer, ultimately affecting tumor growth and metastasis. Recent studies have shed light on the ability of miRNAs to regulate chromatin structure and gene expression by targeting the enzymes involved in DNA methylation and histone modification processes. This emerging understanding of miRNA-mediated epigenetic regulation opens new avenues for breast cancer therapy, emphasizing the need for further research to explore the potential of miRNAs as therapeutic agents or targets [23]. Notably, specific miRNAs (e.g., miR-34a, miR-16) have been linked to senescence and the SASP, suggesting that modulating these miRNAs could delay therapy-induced aging phenotypes [23,24].

**Long Non-coding RNA and Epigenetics:** Similarly to miRNAs, lncRNAs (classified as transcripts longer than 200 nucleotides without coding potential) are also commonly expressed aberrantly in breast cancers and contribute to cancer development, progression, recurrence, and therapeutic resistance. Uniquely, lncRNAs exert their regulatory functions by forming complex secondary and tertiary structures that interact with other transcripts, chromatin, and/or proteins [25]. The literature has identified that prolonged chemotherapeutic treatment leads to lncRNA-mediated chemoresistance through various mechanisms, including multidrug efflux, suppression of apoptosis, DNA damage response, and epigenetic alterations, as well as functioning as competitive endogenous RNA [26]. This phenomenon has led to the failure of achieving long-term remission in patients treated with CDK4/6 inhibitors [27]. In addition, lncRNAs have also been found to have interplay with miRNA, serving as “sponges” or competing endogenous RNAs (ceRNAs), thereby attenuating the repression of mRNAs by miRNAs [28]. Therefore, the complexity of the lncRNA–microRNA axes calls for further characterization and understanding of the non-coding genome and continued study as a new outlook in the therapeutic development strategies for breast cancer.

Moreover, the interplay between non-coding RNAs and epigenetic modulators (e.g., DNMTs and HDACs) can amplify or dampen senescence signals in both tumor and stromal cells, revealing a coordinated regulatory network that underpins therapy-induced aging effects [29].

## 3. Therapeutic Intervention-Induced Adverse Effects and Their Relationship to Aging

Table 1 highlights potential mechanisms associated with therapy-induced accelerated aging and delayed symptoms that they might have developed due to breast cancer treatments.

Many breast cancer therapies have been associated with various clinical manifestations of adverse events. Table 1 highlights the ones that are particularly of interest to accelerated aging and the potential delayed symptoms that they might develop due to breast cancer treatments.

**Cardiovascular and Respiratory Long-term Effects:** Previous reviews have identified that breast cancer patients, both pre and post chemotherapy, have decreased cardiorespiratory fitness (CRF) or Vo2max [24]. Specifically, many chemotherapy agents and radiation have been associated with post-treatment cardiomyopathy. Zambetti et al., in a 14-year follow-up study, found that doxorubicin is associated with cardiac pathology, including systolic dysfunction and decreased left ventricular ejection fraction (LVEF). They also reported that breast irradiation had a significant impact on early congestive heart failure (CHF) [25]. These findings were corroborated by later studies, which reaffirmed that the mean LVEF in the doxorubicin-treated group was statistically significantly lower in the 5- to 8-year sample post chemotherapy [26]. Anthracyclines have also been found to be particularly associated with delayed cardiomyopathy [27]. Women aged 66 to 70 years who received adjuvant anthracyclines had significantly higher rates of CHF, and the difference in rates of CHF continued to increase through more than 10 years of follow-up [28]. While the current understanding of the mechanism of cardiovascular dysfunction is limited, Szczepaniak et al. proposed chemotherapy-induced damage to the vascular endothelium. They found that endothelium-dependent NO (nitric oxide)-mediated vasodilation was severely impaired in patients after neoadjuvant chemotherapy, while endothelium-independent responses remained normal [29]. Lastly, there has been a call to develop a comprehensive stratification method for survivors of breast cancer to predict their risk of major cardiovascular events post breast cancer therapy. The CHEMO-RADIAT (a score that includes congestive heart failure, hypertension, elderly, myocardial infarction/peripheral artery occlusive disease, obesity, renal failure, abnormal lipid profile, diabetes mellitus, irradiation of the left breast, anthracycline dose, and transient ischemic attack/stroke) offers one potential stratification method to aid physicians in multidisciplinary decision making [35]. Emerging research suggests that these cardiotoxic effects are also driven by epigenetic alterations in cardiac myocytes and endothelial cells, which parallel age-related cardiac dysfunction.

**Endocrine System and Reproductive Effects:** A large proportion of women during, or after breast cancer therapy experience menopausal symptoms or clinical manifestation of estrogen deficiency [35]. For instance, premenopausal women can develop ovarian deficiency and infertility, while postmenopausal women treated with aromatase inhibitors (AI) may experience arthralgia, accelerated bone loss, and osteoporotic fractures [36,37,38]. However, the relationship between bone health and breast cancer treatment is intricate and drug-specific. For example, tamoxifen, a selective estrogen receptor modulator (SERM), is known to cause bone formation and has a protective effect on bone health in postmenopausal women. Yet, in younger, premenopausal women, tamoxifen has the opposite effects, resulting in an elevated risk of pathologic fractures [39]. There have been abundant studies attempting to treat these specific therapy-induced early endocrine dysfunctions, especially in premenopausal women, such as temporary ovarian suppression by administering a gonadotropin-releasing hormone analog or luteinizing hormone-releasing hormone agonists (LHRHas) [40,41,42], with non-uniform benefits described. Further, there has been an emphasis on understanding the long-term QoL for women of childbearing age who underwent chemotherapeutic regimens [42]. Notably, certain epigenetic modifications in reproductive tissues—such as the altered histone acetylation of key ovarian genes—have been linked to premature ovarian failure in breast cancer survivors, indicating a direct link between therapy, epigenetics, and endocrine aging [43]. Additionally, new research suggests that therapy-induced epigenetic dysregulation may increase the risk of early metabolic syndrome, especially among survivors who experienced chemotherapy-induced menopause. Interestingly, altered metabolic hormones, including leptin and adiponectin, may themselves undergo DNA methylation changes influenced by physiological states such as obesity. This epigenetic regulation further strengthens the link between endocrine health and breast cancer carcinogenesis [44,45,46,47]. Specifically, elevated leptin levels promote cancer cell proliferation, angiogenesis, invasion, and metastasis by activating multiple signaling pathways, including JAK/STAT, PI3K/AKT, and MAPK/ERK. Furthermore, leptin is involved in the estrogen synthesis pathway, increasing local estrogen production via the upregulation of aromatase expression, thereby exacerbating hormone receptor-positive breast cancer progression [48]. Conversely, adiponectin, another key adipokine, has protective effects against breast cancer by inhibiting proliferation and inducing apoptosis. Lower adiponectin levels, characteristic of obesity, correlate with a larger tumor size and poorer prognosis. Mechanistically, adiponectin exerts its tumor-suppressive effects by activating AMP-activated protein kinase (AMPK), suppressing the PI3K/AKT pathway, and counteracting leptin’s oncogenic actions [48]. This intricate interplay between metabolic state and cancer treatment raises a compelling “chicken or the egg” question, specifically whether metabolic dysregulation drives therapy resistance or if treatment-induced changes exacerbate metabolic dysfunction. This duality presents both opportunities for targeted interventions and challenges in managing the long-term health of breast cancer survivors.

## 4. Targeting Epigenetic Regulation in Breast Cancer Therapy

Understanding the epigenetic mechanisms by which breast cancer treatments impact aging-related pathways opens possibilities for developing therapeutic interventions. Table 2 provides a summary of potential interventions to reverse or mitigate therapy-related accelerated aging. Agents that can reverse or mitigate epigenetic changes, such as DNA methylation inhibitors, HDAC inhibitors, and miRNA mimics or inhibitors, have the potential to protect against treatment-induced aging effects. Furthermore, lifestyle interventions and dietary components that influence the epigenome, such as exercise, caloric restriction, and polyphenol-rich foods, may also offer non-pharmacological strategies to counteract aging processes exacerbated by cancer therapy. Interventions such as metformin therapy and weight loss have not shown significant associations with epigenetic aging in overweight postmenopausal breast cancer survivors [45]. However, identifying epigenetic biomarkers and targeting specific epigenetic modifications offer new avenues for therapy. For instance, screening for H4K12 acetylation and H3K27 acetylation has identified potential therapeutic targets in ER-positive breast cancer [46].

**Epigenetic Therapy in Clinical Trials:** Clinical trials exploring epigenetic therapies, including the use of DNA methyltransferase inhibitors (DNMTis) and histone deacetylase inhibitors (HDACis), have shown potential in treating breast cancer [31,53]. These therapies target epigenetic mechanisms that play a crucial role in tumor progression. DNMTis, such as azacitidine, and HDACis, such as vorinostat, have been used in combination with chemotherapy to treat breast cancer [31,53]. Additionally, epigenetic changes, including DNA methylation and histone acetylation or methylation, have been implicated in the development of breast cancer [54]. The aberrant methylation of gene promoter regions and the dysregulation of histone modifications have been associated with reduced gene function and the evolution of human cancer. The reversibility of epigenetic changes makes them attractive targets for breast cancer therapy. Moreover, ongoing studies are investigating next-generation epigenetic drugs (e.g., LSD1 inhibitors and BET inhibitors) in combination with endocrine therapy to address both tumor progression and therapy-related aging [30,32,33]. Among these emerging agents, combined LSD1(lysine-specific histone demethylase 1) and HDAC inhibition has shown promise in pre-clinical models of triple-negative breast cancer, reducing proliferative capacity while simultaneously decreasing senescence-associated inflammation. This dual effect may help extend both overall survival and health span in breast cancer survivors [34,49,50].

**Induced Senescence as a Therapeutic Target:** The research conducted by Zhu et al. shed light on the role of epigenetic therapy utilizing the mechanism of cancer cell senescence [31]. Specifically, their study focused on the co-targeting of CDK4/6 and BRD4, which halted cancer cell proliferation, induced cellular senescence, and enhanced sensitivity to ferroptosis through epigenetic modifications. This intriguing finding suggests that epigenetic alterations are vital in mediating therapeutic responses and may accelerate aging processes in cancer cells. These findings have significant implications for developing novel therapeutic strategies that target epigenetic regulators to improve treatment outcomes and mitigate the aging-related effects of cancer therapy. It is now recognized that senescence induction can be a double-edged sword. While it can halt tumor cell proliferation, the buildup of senescent cells in tissues can lead to chronic inflammation and impair normal tissue function. Hence, strategies that not only induce tumor cell senescence but also target or remove senescent cells (senolytics) are gaining attention. Combining senolytics with epigenetic modulators could potentially minimize therapy-induced aging while maintaining anti-tumor efficacy [52,55,56]. 

**Epigenetic Biomarkers Impacting Therapeutic Efficacy:** The study conducted by Yip et al. brought attention to the importance of p21, an epigenetic regulator, in drug resistance and therapeutic prioritization for PIK3CA-mutant breast cancer [53]. Their findings suggest that p21, a known influencer of aging processes, could be a biomarker for predicting therapeutic response and gaining insights into the aging landscape in cancer survivors. This highlights the crucial role of epigenetic regulators in determining the efficacy of therapy and understanding the complex interplay between epigenetics and aging in the context of breast cancer. Likewise, the identification of the phosphorylated RB (retinoblastoma) protein and its epigenetic control has emerged as another biomarker for predicting CDK4/6 inhibitor response. This biomarker can help clinicians tailor therapy to minimize overtreatment and potentially spare patients from interventions that might accelerate aging without substantial survival benefits. 

**Dietary and Microbiotic Influences on Epigenetics and Aging:** In a comprehensive review conducted by Wu, the intricate relationship between dietary bioactive compounds, gut microbiota, epigenetic regulation, and aging was explored [51]. This review provided valuable insights into non-pharmacological interventions that may mitigate aging processes and improve therapeutic outcomes in breast cancer survivors. By modulating the gut microbiota and influencing epigenetic regulation, dietary bioactive compounds have the potential to impact the aging landscape in cancer survivors positively. These findings not only open up new avenues for therapeutic interventions but also emphasize the importance of considering the role of diet and microbiota in epigenetics and aging. Future work evaluating the synergy between epigenetic-targeted agents and dietary interventions (e.g., high-fiber diets, probiotic supplementation) will be critical to optimizing survivorship care in older breast cancer patients [57,58,59,60].

Additionally, certain dietary interventions like intermittent fasting or time-restricted feeding may impact the expression of sirtuins (SIRT1–SIRT7), which are histone deacetylases involved in both metabolic regulation and aging processes. Preliminary evidence suggests that modulating sirtuin activity in conjunction with standard therapies could improve metabolic health and reduce the senescence burden, although more clinical trials are required to validate these approaches [61,62,63,64].

## 5. Materials and Methods

In conducting this scoping review, we followed the methodological framework proposed by Arksey and O’Malley (2005) and further refined by the Joanna Briggs Institute guidelines on scoping reviews [65,66]. The primary objective was to map the key concepts relating to “epigenetics”, “aging”, and “breast cancer therapy”, thereby clarifying existing knowledge gaps and identifying potential areas for future research.

## 6. Search Strategy

We developed a systematic search strategy in consultation with a medical librarian. The following databases were queried to identify peer-reviewed articles published between 2010 and 2024: PubMed, Web of Science, Embase, and Scopus. The search terms included MeSH headings and keywords relevant to breast cancer, epigenetics, aging, accelerated senescence, DNA methylation, histone modification, non-coding RNAs, and cancer therapies. Boolean operators (AND, OR) were employed to expand or narrow the search as needed. Only articles in English were considered, although we included references from systematic reviews and meta-analyses that contained relevant non-English studies.

## 7. Screening and Selection of Articles

Two reviewers (NN and ZS) independently screened titles and abstracts using the following predefined inclusion criteria: (1) studies focusing on breast cancer therapies, (2) mention of epigenetic mechanisms, and (3) relevance to aging, accelerated senescence, or related biomarkers in breast cancer survivors. Studies that solely examined risk factors for breast cancer without mentioning epigenetics or aging were excluded, as well as those focusing exclusively on pediatric populations or basic research unrelated to clinical outcomes. Full-text articles were then assessed for eligibility, with disagreements resolved through discussion with a third reviewer (SS).

## 8. Data Charting and Synthesis

A standardized data charting form was utilized to extract key information from each eligible article, including study design, population characteristics, therapeutic regimens, epigenetic endpoints (e.g., DNA methylation patterns, histone modifications, non-coding RNA alterations), and measures of aging (e.g., chronological age, epigenetic clocks, telomere length, incidence of age-related comorbidities). Data were then synthesized thematically, allowing us to categorize findings into sections on DNA methylation, histone modifications, telomere dynamics, chronic inflammation, and non-coding RNA-mediated mechanisms. Through iterative discussions, our team consolidated overlaps and distinctions across studies, focusing on how cancer therapies may induce epigenetic shifts which contribute to aging processes in breast cancer survivors.

## 9. Quality Appraisal

While scoping reviews typically do not require a formal quality assessment of included studies, we conducted a brief methodological appraisal to ensure that studies presented sufficient details on patient populations, epigenetic endpoints, and outcomes related to aging. We discussed the risk of bias and limitations of each study in roundtable sessions, but excluded no studies strictly on quality grounds, consistent with the purpose of a scoping review to comprehensively map the evidence base.

## 10. Analysis of Gaps and Future Directions

Upon completing the thematic analysis, we identified several gaps in the literature: inconsistent use of epigenetic aging biomarkers, limited longitudinal studies that track epigenetic changes over years post therapy, and a paucity of randomized controlled trials incorporating epigenetic endpoints. These gaps highlight the need for more robust, long-term investigations to elucidate whether epigenetic changes are reversible and how intervention strategies—pharmacological or lifestyle-based ones—might mitigate therapy-induced accelerated aging.

By following this scoping review methodology, we aimed to provide an extensive overview of the current state of knowledge on epigenetic modifications associated with breast cancer therapies and their relationship to aging, thereby laying the groundwork for subsequent systematic reviews and meta-analyses which can address more specific research questions.

## 11. Conclusions

Breast cancer therapies contribute to accelerated aging through diverse epigenetic mechanisms, including DNA methylation and histone modifications. Understanding these changes offers insights into the long-term consequences of cancer treatment and highlights the potential for epigenetic-based interventions to mitigate aging effects. Further research is needed to elucidate the full spectrum of epigenetic alterations induced by breast cancer therapies and their impact on patient health span and quality of life.

Ultimately, personalized approaches that integrate both pharmacological and lifestyle-based epigenetic interventions may pave the way for reducing therapy-induced aging and improving survivorship outcomes. In this context, ongoing scientific collaborations among oncologists, geriatricians, and molecular biologists will be essential for translating epigenetic discoveries into clinically meaningful improvements for breast cancer survivors, particularly those at the highest risk of age-related comorbidities.

## Figures and Tables

**Table 1 cancers-17-00866-t001:** Overview of selected breast cancer therapies and their potential epigenetic impacts on aging.

Therapy Type	Mechanism	Potential Epigenetic Changes	Implications for Aging	Key References
Chemotherapy (e.g., anthracyclines)	Induces DNA damage in rapidly dividing cells	Altered DNA methylation profiles and increased histone modifications	Accelerated senescence, SASP induction, and increased inflammatory milieu	[14,18,30]
Radiation	Ionizing radiation causing DNA breaks	Methylation changes and activation of DNA damage response elements	Persistent senescent cell population and higher risk of secondary malignancies	[21,22]
Hormonal therapy (AI, SERMs)	Modulates estrogen receptor signaling	Epigenetic modulation of hormone-responsive genes	Bone density changes, altered endocrine function, and potential reproductive aging	[17,18,19,20]
Targeted therapy (CDK4/6 inhibitors)	Inhibits specific cell cycle regulators (CDK4/6)	Histone acetylation changes via crosstalk with cell cycle regulation	Therapy-induced senescence and potential epigenetic-based resistance	[14,31]
Epigenetic drugs (DNMTis, HDACis, and BETis)	Directly target epigenetic regulators (DNA methylation, histone deacetylation, and bromodomains)	Reactivation of silenced tumor suppressor genes and global histone modifications	Potential reversal of therapy-induced aging and synergy with standard treatments	[24,25,30,32]
Immunotherapy (e.g., checkpoint inhibitors)	Enhances T-cell mediated anti-tumor response	Possible modifications in immune-related gene promoters and histone marks	Impact on immune senescence, interplay with chronic inflammation, and T-cell exhaustion	[33,34]

**Table 2 cancers-17-00866-t002:** Potential epigenetic-based strategies to mitigate therapy-induced aging.

Strategy	Mechanism of Action	Example/Intervention	Expected Outcome	Key References
Pharmacological Epigenetic Modulation	Inhibiting DNMT or HDAC to restore normal gene expression and reduce senescence	Azacitidine (DNMTi), Vorinostat (HDACi), and BET inhibitors	Reactivation of tumor suppressor genes and decreased pro-aging factors	[15,24,30,32]
Targeting Non-coding RNAs	Altering expression of specific miRNAs or lncRNAs to normalize epigenetic profiles	miRNA mimics/inhibitors and lncRNA modulators	Reduced therapy resistance, decreased inflammation, and less senescence	[23,34,49]
Lifestyle and Nutritional Interventions	Dietary changes and physical exercise to modulate the epigenome	Polyphenol-rich diet, calorie restriction, and aerobic exercise	Decreased SASP factors, improved DNA repair, and overall anti-aging effects	[27,45,50,51]
Combination Therapies	Concurrent use of epigenetic drugs with standard chemotherapy or hormonal therapy	DNMTi + chemotherapy or HDACi + endocrine therapy	Enhanced treatment efficacy and protection against therapy-induced aging	[25,31,52]
Cardioprotective Agents	Mitigating epigenetically driven cardiotoxic effects of breast cancer treatments	ACE inhibitors and beta-blockers	Reduced therapy-induced endothelial and cardiomyocyte senescence	[29,35,50]

## Data Availability

No new data were created or analyzed in this study. Data sharing is not applicable to this article.

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
