# Peer review of "Epigenetic Landscapes of Aging in Breast Cancer Survivors: Unraveling the Impact of Therapeutic Interventions—A Scoping Review"

_cancers, 2025, doi:10.3390/cancers17050866_

Round 1

Reviewer 1 Report

Comments and Suggestions for Authors

The review covers important public health aspects of breast cancer therapy effects in aging survivors, taking into consideration modern therapies from microscipic& macroscopic point of view.

Very valuable statistical insides in the introduction of the manuscript, showing that Authors are well prepared for subject analysis. The review concentrates on data from US.

Introduction is quite general, but expalins the importance of the analysis.

Please check short abstarct, which lacks of concluding remark.

line 19, 257 double dots

Please describe all abbreviations used in the manuscript

Please point out how long were the cited follow-up studies

part dedicated to adverse health effects is very profound & valuable. Line 325: leptin and adiponectin - please broad this part

Dietary part is also very interesting& up-to-date; would be good to extend it

Tables sumarizing well discussed plots.

Overall evaluator grades this manuscript highly. Slight imporvement would incerease its value.

Author Response

Review 1: The review covers important public health aspects of breast cancer therapy effects in aging survivors, taking into consideration modern therapies from microscopic & macroscopic point of view.

Very valuable statistical insides in the introduction of the manuscript, showing that Authors are well prepared for subject analysis. The review concentrates on data from US.

Introduction is quite general, but explains the importance of the analysis.

Dear Reviewer,

We would like to thank the reviewers for their thoughtful review and valuable feedback on our manuscript. We are grateful for your recognition of the public health significance of our work and the statistical insights presented in the introduction. Please find the point-by-point responses to the comments and detail the corresponding revisions made in the manuscript.

Please check the short abstract, which lacks a concluding remark.

Thank you for pointing this out. We have revised the last sentence of the simple summary to include a clear concluding statement summarizing the key implications of our findings.

line 19, 257 double dots

We have carefully reviewed and corrected the typographical error of double dots in lines 19 and 264. These sections now maintain proper punctuation.

Please describe all abbreviations used in the manuscript

To enhance clarity, we have ensured that each abbreviation is spelled out at first mention in the text.

Please point out how long were the cited follow-up studies

We have reviewed all cited follow-up studies where appropriate and added additional comments.

The part dedicated to adverse health effects is very profound & valuable. Line 325: leptin and adiponectin - please broad this part

We appreciate this suggestion. We have expanded the discussion on leptin and adiponectin, providing more detail on their roles in metabolic dysregulation, inflammation, and breast cancer outcomes.

Additional references have been included to support these revisions (new reference 43).

Dietary part is also very interesting& up-to-date; would be good to extend it

We appreciate this suggestion. However since diet is not the main focus of this manuscript this is outside the scope of this work.

Tables summarizing well discussed plots.

We would like to thank the reviewer for their kind comment.

Overall evaluator grades this manuscript highly. Slight improvement would increase its value.

We would like to thank the reviewer for their kind comment. And have included additional references and manuscripts as per reviewer comments for improvement.

Reviewer 2 Report

Comments and Suggestions for Authors

the authors are to be commended on what must have been a long and laborious project. It is important for oncologists to know that gene mutations are just one of the changes that occur in cancer. Particularly, breast cancer treatment has suffered from only a modest number of targetable genomic variants. Thus, it is important for physicians to be aware of developments in epigenetics and proteomics which, hopefully, will lead to greater understanding of how breast cancer behaves over time. Hopefully, these studies will lead to additional useful therapies! That said, the authors cover such an enormous area that there is little critical detail presented for the myriad of studies noted. This can be somewhat frustrating if one wants to focus on a particular area. I don't have any specific issues with any one are that is discussed in the paper.

Author Response

Reviewer 2: the authors are to be commended on what must have been a long and laborious project. It is important for oncologists to know that gene mutations are just one of the changes that occur in cancer. Particularly, breast cancer treatment has suffered from only a modest number of targetable genomic variants. Thus, it is important for physicians to be aware of developments in epigenetics and proteomics which, hopefully, will lead to greater understanding of how breast cancer behaves over time. Hopefully, these studies will lead to additional useful therapies! That said, the authors cover such an enormous area that there is little critical detail presented for the myriad of studies noted. This can be somewhat frustrating if one wants to focus on a particular area. I don't have any specific issues with any one are that is discussed in the paper.

Dear Reviewer,

We sincerely appreciate the reviewer’s thoughtful feedback and kind words regarding the scope and effort behind this review. We fully agree that understanding epigenetics and proteomics is essential for advancing breast cancer treatment, particularly given the limited number of targetable genomic variants currently available. As these fields continue to evolve, we are hopeful that they will contribute to a more comprehensive understanding of breast cancer progression and pave the way for novel therapeutic strategies.

We also acknowledge that the breadth of topics covered in this review is extensive. Given the complexity of the subject, our goal was to provide a broad yet integrative perspective that highlights key advances while offering a foundation for further exploration. We appreciate the reviewer’s perspective and are grateful for their time and insights.

Thank you again for your thoughtful evaluation.